# Do the Results of Bile Cultures Affect the Outcomes of Patients with Mild-to-Moderate Ascending Cholangitis? A Single Center Prospective Study

**DOI:** 10.3390/diagnostics15060695

**Published:** 2025-03-11

**Authors:** Yoav Krupik, Eran Ariam, Daniel L. Cohen, Anton Bermont, Sergei Vosko, Haim Shirin, Shay Matalon

**Affiliations:** Gonczarowski Family Institute of Gastroenterology and Liver Diseases, Shamir (Assaf Harofeh) Medical Center, Faculty of Medicine, Tel Aviv University, Zerifin 6997801, Israel; eranariam@gmail.com (E.A.); bermont@doctor.com (A.B.); sergeivo@shamir.gov.il (S.V.); haimsh@shamir.gov.il (H.S.); matalons@shamir.gov.il (S.M.)

**Keywords:** bile, culture, antibiotics, ERCP

## Abstract

**Background/Objectives:** Bile cultures are recommended in patients undergoing endoscopic retrograde cholangiopancreatography (ERCP). We sought to evaluate if bile cultures affect the outcomes of patients with mild-to-moderate ascending cholangitis. **Methods:** Bile cultures were prospectively obtained from patients undergoing ERCP between 2021 and 2023 at a single gastroenterology unit. The primary outcome was the prognosis of patients with mild-to-moderate ascending cholangitis who received appropriate antibiotic coverage with empiric antibiotics versus those with bacteria resistant to the empiric antibiotics. Additionally, outcomes between those with positive and negative biliary cultures were assessed. **Results:** One hundred sixty ERCPs were conducted, including 65 (40%) for ascending cholangitis with a naive papilla. Of these, 43 (66.2%) had a positive bile culture. Fourteen (32.6%) described mixed bacteria. *Enterococcus* spp. was the most common bacteria (22, 51.2%), followed by *E. coli* (17, 39.6%). Patients that were treated with appropriate antibiotics had similar outcomes compared to those who received inappropriate antibiotics per bile culture susceptibilities in terms of length of hospitalization (7.8 days vs. 7.9 days), in-hospital mortality, 30-day readmissions, and 30-day mortality (*p* ≥ 0.21, ns). There were also no differences in those outcomes between patients with positive and negative bile cultures (*p* ≥ 0.09, ns). **Conclusions:** These results question the need for obtaining bile cultures in every ERCP performed, including those with cholangitis. They imply that decompression of the biliary tree during ERCP is the more significant aspect of treatment, rather than the selection of an appropriate antibiotic regimen. Additional studies are needed to assess the benefits of acquiring bile cultures in all cases of ascending cholangitis.

## 1. Introduction

In healthy individuals, bile is basically sterile [1]. There are several conditions in which pathogens may contaminate bile, such as obstruction of the bile ducts by stones or malignancy [2,3] or bile duct manipulations such as endoscopic retrograde cholangiopancreatography (ERCP) and papillotomy [4]. The combination of bacteria in the bile along with biliary obstruction may lead to acute ascending cholangitis, a life-threatening condition in which prompt restoration of the bile flow via ERCP and antibiotics is mandatory [5,6,7,8,9].

According to the Tokyo guidelines, bile cultures should be obtained in every case of ascending cholangitis in order to identify the causative organism, while blood cultures are recommended only in severe cases of acute cholangitis [10]. The most common pathogens identified in patients with acute cholangitis are *Escherichia coli*, *Klebsiella* spp., *Enterococcus* spp., *Pseudomonas* spp., and *Enterobacter* spp. [4,11,12,13]. Antimicrobial therapy guidelines are based on the pathogens’ susceptibility [10]. With the widespread use of antibiotics and the global emergence of multidrug-resistant organisms, efforts are underway to identify microbiological characteristics and patterns of drug resistance related to intra-abdominal infections. These will potentially help in directing antimicrobial choice, allowing early de-escalation to pathogen-directed therapy, and the appropriate termination of therapy. Prior studies and new recommendations from medical societies reinforce these principles [13,14,15,16,17]. For example, a recent study by George et al. has concluded that bile cultures may guide antibiotic treatment for ascending cholangitis based on the significantly higher rate of positive bile cultures compared to blood cultures, with 84.4% versus 35% of the cases, respectively. However, that study did not describe the effect of directed treatment on patient outcome measures [18].

On the basis of the recommendation that bile cultures be taken from all cases of ascending cholangitis for antibiotic susceptibility, we aimed to prospectively assess whether this practice benefitted patients in terms of improved outcomes. Secondarily, we sought to describe the bacterial profile of patients with ascending cholangitis.

## 2. Materials and Methods

### 2.1. Study Design

This was a single-center, prospective study performed between 2021 and 2023 at Shamir Medical Center. The study was approved by the medical center’s Institutional Review Board (approval #174-21-ASF). All authors had access to the study data and reviewed and approved the final manuscript.

All adult patients undergoing ERCP were prospectively recruited for the study. During ERCP, bile was aspirated from all participants and sent for culture and antibiotic susceptibility. While all patients undergoing ERCP were included in the study, the main focus was on patients with ascending cholangitis. Cholangitis was defined by the Tokyo guidelines as the combination of systemic inflammation, serum blood tests suggestive of cholestasis, and imaging that support the presence of bile duct obstruction [10]. Patients with mild-to-moderate ascending cholangitis were included in the study. Patients with severe cholangitis were excluded due to challenges in obtaining informed consent and the potential confounding effect of urgent care.

### 2.2. Patients and Outcomes

All adult patients (age 18 years old or greater) who provided informed consent were included. Patient data included blood test results, age, ERCP indication, prior ERCP, prior sphincterotomy, blood and bile culture results, bacterial susceptibility, ERCP sphincterotomy, hospitalization information including antibiotics provided to the patient, length of hospitalization, need for ICU care, intubation, 30-day mortality, and rehospitalization rates.

The assessed outcomes included identifying bacterial resistance profiles in bile cultures, checking the correlation between blood and bile cultures, and assessing the influence of the bacteria identified in the cultures on the prognosis of the patients with outcomes including the length of hospitalization and in-hospital mortality.

Additionally, the local bile microbiome of all patients who underwent ERCP was described.

Patients were defined as having received appropriate empiric antibiotics if the bacterial susceptibility of the bile culture was covered by the antibiotics that the patient was receiving. The antibiotic coverage was defined as inappropriate if the bile susceptibility profile showed resistance to the empiric antibiotics that the patient was receiving.

### 2.3. Statistical Analysis

Data were analyzed using RStudio software version 4.3.2 (RStudio, Inc., Boston, MA, USA). Numerical variables are given as numbers. Categorical variables are given as numbers and percents. Welch’s test was used to test for statistical significance among numerical variables. Fisher’s exact test was used to test for significance among categorical variables. Finally, a multivariate logistic regression model was used to assess factors associated with positive bile cultures.

All *p* values were 2-sided, and a *p* value of <0.05 was considered statistically significant.

## 3. Results

### 3.1. The Overall Study Population

During the study period, 160 ERCPs with bile aspirates were performed. A flowchart of the study patients can be seen in Figure 1. Of the 160 total patients, 71 patients (44.4%) were diagnosed with ascending cholangitis. Of these cholangitis patients, 65 (91.5%) had a naïve papilla (no prior sphincterotomy). Forty-seven (72%) patients were diagnosed with moderate cholangitis according to the Tokyo guidelines, while 18 (28%) were mild cases. Of the 65 patients with a naïve papilla, 43 (66.2%) had a positive bile culture. Based on the antibiotic susceptibility of the bile cultures, 17 of the 43 (39.5%) were considered to have been treated with appropriate empiric antibiotic treatment, while 26 (60.5%) had been treated inappropriately.

Additionally, 89 (55%) patients underwent ERCP due to indications other than cholangitis, including 25 (28%) due to stent extraction, 15 (17%) due to evaluation of abnormal narrowing of the CBD on imaging (malignant or otherwise), and 49 (55%) due to choledocholithiasis. Of the 64 total patients with either abnormal CBD narrowing on imaging or choledocholithiasis, 51 (80%) had not previously undergone an ERCP or other biliary tree manipulation. Of these 51 patients with a naïve papilla, 25 had a positive bile culture (49.0%).

### 3.2. Patients with a Naïve Papilla Diagnosed with Ascending Cholangitis

The 65 patients with ascending cholangitis and a naïve papilla were evaluated separately and are described in Table 1. The median age was 75 years (interquartile range 60–84.5), and 38 (58%) were female. Only 5 (7.9%) of the patients were nursing home residents, and 14 (21.5%) had a previous hospitalization within a year of the index ERCP. Twenty-two (33.8%) of the patients had diabetes mellitus, 6 (9.2%) had cancer, 2 (3.1%) had cirrhosis, and 1 (1.5%) was receiving immunosuppression.

The mean length of hospitalization was 7.2 days (median—6 days). In addition, 93.8% of patients received antibiotics upon hospitalization. ERCP was performed after a median of 3 days of hospitalization (assessed conservatively by counting calendar days from the date of admission until the day of the procedure; thus, the procedure was in fact performed in most cases within 48 h). The combination of ceftriaxone and metronidazole was the most common empiric regimen prescribed for patients (64.6%). Sphincterotomy was performed in 80% of the cases. The mean width of the common bile duct was 1.29 cm. In-hospital mortality was low, with only one patient (1.5%) dying due to septic shock that worsened after the procedure despite successful biliary decompression with a stent. Ten patients (15%) were re-admitted to the hospital within 30 days of the procedure for reasons unrelated to the original diagnosis of cholangitis. One of them died due to complications from metastatic cancer.

### 3.3. Results of Bile Cultures in Ascending Cholangitis Patients

Overall, 43 of the 65 (66.2%) of the patients with a naïve papilla diagnosed with cholangitis had a positive bile culture, and 15 of them (23%) had positive blood cultures. Of those with positive blood cultures, only 9 (60%) had the same bacteria as found in the bile. Fourteen (32.6%) of the bile cultures described mixed bacteria. Of the specific bacteria found, *Enterococcus* spp. was the most common bacteria found in bile cultures (22, 51.2%) with *E. coli* being the second most common (17, 39.6%). Blood and bile isolates are described in Table 2 for both patients with and without cholangitis.

Bacterial resistance in patients with cholangitis is described in Table 3. Out of the total number of bacteria isolated from bile aspirates, 43% were resistant to ampicillin (27.2% of *Enterococcus* spp.) and 21.3% had ESBL (46.6% of *Enterobacteriaceae* species). Additionally, 23.0% were resistant to ciprofloxacin and 6.1% were resistant to levofloxacin. Finally, 1.5% of bacteria isolated were resistant to vancomycin (4.5% of *Enterococcus* spp.).

### 3.4. Outcomes of Cholangitis Patients Based on Bile Culture Results

A comparison between cholangitis patients with positive bile cultures and those with negative bile cultures was performed (Table 4). Patients with positive cultures were significantly older (80 vs. 67.5, *p* = 0.02) and had a higher percentage of moderate cholangitis (81% vs. 55%, *p* = 0.03). They also had more diabetes mellitus (44% vs. 13.3%, *p* = 0.01). However, in a multivariate logistic regression analysis including age, diabetes, and cholangitis severity (Table 5), none of these factors remained significantly associated with positive bile cultures. This may be due to confounding variables, collinearity, and small sample size.

Patients with a positive bile culture had a mean hospitalization of 7.8 days, while patients with a negative culture had an average of 5.9 days (*p* = 0.09). Six out of the 10 patients who were re-hospitalized had a positive bile culture, while 4 had a negative culture (*p* = 0.72). One patient with a negative culture died during the 30-day follow-up period of complications from metastatic cancer (*p* = 0.33).

Additional comparisons were performed between those with positive bile culture susceptibility showing appropriate empiric antibiotics and those with inappropriate antibiotics (Table 6). Out of the patients with a positive bile culture, those with an appropriate empirical antibiotic regimen had an average hospital stay of 7.8 days, while those who were treated with an inappropriate antibiotic regimen stayed at the hospital for 7.9 days on average (*p* value = 0.95).

Out of the six patients that were re-hospitalized after having had a positive bile culture during the initial hospitalization, three had been treated appropriately and three had been treated inappropriately (*p* = 0.57).

## 4. Discussion

Our study prospectively evaluated the possible benefits of obtaining bile cultures for the management and prognosis of patients with acute cholangitis with a naïve papilla. We found that there was no significant difference regarding patients’ prognosis between patients treated with an empirical antibiotic regimen that suited the bile bacterial susceptibility compared to those treated “inappropriately”, bringing into question the benefits of antibiotic susceptibility testing. Additionally, outcomes of patients with positive bile cultures compared to those with negative cultures were similar, although there was a trend towards fewer hospitalization days in the culture-negative group (mean 7.8 vs. mean 5.9, *p* value = 0.09) as expected.

Culture-based therapy continues to be the gold standard for many infections, including ascending cholangitis, given the risk of inappropriate treatment with empiric antibiotics. Indeed, the Tokyo consensus guidelines recommend that bile cultures should be obtained at the beginning of any procedure performed for the indication of ascending cholangitis, even though historically there has been no high-quality evidence to support this statement. Thus, the Tokyo guidelines present this statement as “strong recommendation, very low quality of evidence” [10,19].

Similar to our study, this assertion was challenged in the past by Masuda et al., who also found no notable distinctions in the duration of hospitalization, in-hospital mortality, and increased disease severity rates between patients treated for cholangitis with initial antibiotic-resistant cultures and those with initial antibiotic-sensitive cultures. The only significant difference observed in their study was a higher incidence of post-ERCP cholecystitis in the antibiotic-resistant group compared to the antibiotic-sensitive group (*p* = 0.0245) [20].

Our study is the first prospective study to present an insight into the bile flora profile of patients diagnosed with ascending cholangitis in Israel. According to the Tokyo guidelines, the most frequent organism found in the bile of patients diagnosed with ascending cholangitis is *E. coli* (31–44%), followed by *Klebsiella* spp. (9–20%) and *Enterococcus* spp. (3–34%) [10]. Unlike in most studies cited in the guidelines, the most frequent organism isolated from bile in our study was *Enterococcus* spp. and not *E. coli* or other Gram-negative bacteria. A study by Gromski et al. also showed similar results to ours [13]. Of note, the most common bacteria found in the blood of the patients with bacteremia was *E. coli*, a fact that could strengthen past findings regarding the pathogenicity of enterococci in intra-abdominal infections [21].

A unique analysis performed by Forster et al. isolated bacterial cultures from bile aspirated during ERCP in patients with indications unrelated to cholangitis (e.g., post-liver transplant strictures, primary sclerosing cholangitis, etc.), assuming bile sterility. Their results revealed *Enterococcus* spp. as the predominant bacteria (67%), followed by *E. coli* (32.2%) and *Klebsiella* (28.2%). These proportions align more closely with our findings, particularly when compared to our sub-cohort of 51 patients with a naïve papilla and without cholangitis, where we observed 44% positivity for *Enterococcus* spp. and 20% for *E. coli* and *Klebsiella* spp. each [22].

Our study also found a large proportion of cholangitis patients with naïve papilla positive for mixed bacteria in their bile aspirates (32.63%), as well as in aspirates from patients with a naïve papilla that were treated due to other indications and did not meet the criteria for cholangitis (28%). These numbers, along with an only 60% correlation between positive bile and blood bacteria, suggest that aspirating bile during ERCP is inherently not a sterile procedure, and careful clinical judgment should be applied while choosing antibiotic treatment.

Bacterial resistance, and ESBL prevalence in particular, has also been studied and documented in the past. For instance, Goo et al. found that resistant bacteria made up 7.4% of blood and 13.6% of bile isolates from patients with acute cholangitis [23]. Melzer et al. found 18% ESBL bacteria out of all Gram-negative isolates from bile [24]. Additionally, Miutescu et al. observed a presence of 8.9% ESBL bacteria in patients with a naive papilla in comparison to 28.7% in patients that have undergone a previous sphincterotomy and stent placement [25]. The percent of resistant bacteria in bile in our patients with ascending cholangitis was higher in comparison to other studies, including 25% with ESBL bacteria. However, our findings of a high prevalence of ESBL bacteria are consistent with previous data regarding the general prevalence of resistant organisms in the Israeli population [26,27].

Our study has some limitations. Firstly, a relatively small number of patients from a single center were included. Secondly, only patients with mild-to-moderate cholangitis were included in the study. This decision was made to uphold ethical standards, as obtaining informed consent in critically ill patients can be challenging and may delay urgent interventions. Additionally, severe cases often require immediate broad-spectrum empiric antibiotics and early drainage, making bile culture results less influential in guiding therapy. This fact may have led to an underestimation of morbidity and mortality outcomes as sicker patients were not included. Finally, the length of hospitalization was affected by factors unrelated to the episode of cholangitis itself or direct ERCP complications, such as hospital-acquired respiratory infections, oncologic follow-up and treatment, and waiting for a nursing home placement.

These results, while taking into account the study’s limitations and the need for further investigation, raise the question of the significance of obtaining bile cultures in every ERCP performed, including those with ascending cholangitis. Our results imply that decompression of the biliary tree during ERCP is the more significant aspect of treatment, rather than the selection of an appropriate antibiotic regimen. A recent study by Srinu et al., for example, has shown that a short duration of antibiotics is noninferior to the conventional duration in the treatment of moderate to severe cholangitis, a finding that also suggests that the role of antibiotics is limited in these cases [28]. These assertions are consistent with the concept of “source control”, which includes procedures such as the drainage of the abscesses and debridement of necrotic tissue to fight infections, guiding physicians for thousands of years since the days of ancient Egypt [29]. In addition to the fact that acquiring bile cultures from every patient with ascending cholangitis is time-consuming for the team performing the ERCP procedures and the hospital’s laboratory, it also increases costs and prolongs the procedure, potentially putting patients at further risk.

## 5. Conclusions

In conclusion, we have found a high prevalence of *Enterococcus* spp. and *ESBL Enterobacteriaceae* among the patients with ascending cholangitis included in our study, often not covered by standard antibiotics. Nevertheless, the choice of antibiotics did not significantly affect patient outcomes, including in those with resistant bacteria. More research with larger cohorts is needed to clarify the benefit of routine bile cultures in all ERCP-treated patients, as recommended by the Tokyo guidelines.

## Figures and Tables

**Figure 1 diagnostics-15-00695-f001:**
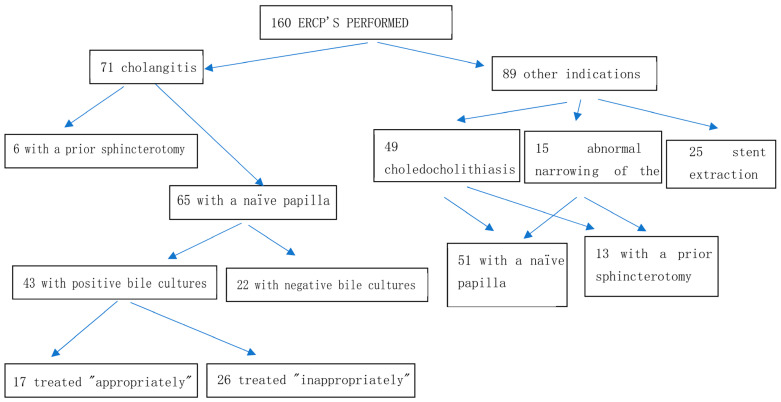
Flowchart of the study patients.

**Table 1 diagnostics-15-00695-t001:** Characteristics of patients with cholangitis and a naïve papilla undergoing ERCP.

	*n* = 65
**Age (years median)**	75
**Female**	38 (58%)
**Moderate cholangitis**	47 (72%)
**Mild cholangitis**	18 (28%)
**Days before ERCP (median)**	3
**Hospitalization in previous year**	14 (21.5%)
**Resident in nursing home**	5 (7.7%)
**Diabetes mellitus**	2 (33.8%)
**Cancer**	6 (9.2%)
**Cirrhosis**	2 (3%)
**Hemodialysis**	0 (0%)
**Immunosuppression**	1 (1.5%)
**Received antibiotics before ERCP**	61 (93.8%)
**Antibiotics received**	
**Ceftriaxone**	10 (15.3%)
**Metronidazole**	4 (6.1%)
**Ciprofloxacin**	1 (1.5%)
**Piperacillin/tazobactam**	1 (1.5%)
**Meropenem**	2 (3.1%)
**Ceftriaxone + metronidazole**	42 (64.6%)
**Ciprofloxacin + metronidazole**	1 (1.5%)
**Sphincterotomy performed during ERCP**	52 (80%)
**Common bile duct diameter (mean, cm)**	1.29
**Patient outcomes**	
**Length of Hospitalization (days, mean/median)**	7.2/6
**In-hospital mortality**	1 (1.5%)
**Required Intubation**	1 (1.5%)
**Loss of consciousness**	1 (1.5%)
**Hypotension requiring vasopressors**	1 (1.5%)
**30-day re-hospitalizations**	10 (15.3%)
**30-day mortality**	1 (1.5%)

**Table 2 diagnostics-15-00695-t002:** Blood and bile culture isolates.

Patients with Naïve Papilla	Cholangitis	Without Cholangitis
**Positive bile culture**	43 (66.1%)	25 (49%)
**Bacteria**		
**Mixed bacteria**	14 (32.6%)	7 (28%)
** *E. coli* **	17 (39.6%)	5 (20%)
***Klebsiela* spp.**	9 (21%)	5 (20%)
***Pseudomonas* spp.**	3 (7%)	0 (0%)
***Enterococcus* spp.**	22 (51.2%)	11 (44%)
***Acinetobacter* spp.**	0 (0%)	0 (0%)
**Others**	8 (18.6%)	9 (36%)
**Positive blood culture**	15 (23%)	7 (18.4%)
**Bacteria**		
** *E. coli* **	8 (53.5%)	0 (0%)
***Klebsiela* spp.**	4 (26.7%)	1 (14.1%)
***Pseudomonas* spp.**	0 (0%)	0 (0%)
***Enterococcus* spp.**	5 (33.4%)	3 (42.8%)
***Acinetobacter* spp.**	0 (0%)	0 (0%)
**Others**	2 (13.3%)	3 (42.8%)

**Table 3 diagnostics-15-00695-t003:** Resistance profile in cholangitis with a positive bile culture.

**Ampicillin**	43%
**ESBL**	21.53%
**Ciprofloxacin**	23.07%
**Levofloxacin**	6.15%
**Carbapenem**	1.53%
**Macrolides**	12.3%
**Aminoglycosides**	10.7%
**Vancomycin**	1.5%

**Table 4 diagnostics-15-00695-t004:** Comparison of characteristics and outcomes of patients with cholangitis with positive and negative bile cultures.

	Positive Bile Culture *n* = 43	Negative Bile Culture *n* = 22	*p*-Value
**Age (years median)**	80	67.5	0.02
**Female**	26 (60%)	12 (54.5%)	0.79
**Severity of cholangitis**			
**Moderate cholangitis**	35 (81%)	12 (55%)	0.03
**Mild cholangitis**	8 (19%)	10 (45%)	
**Days before ERCP (median)**	3	3	0.31
**Received antibiotics before ERCP**	40 (90%)	19 (86.3%)	0.39
**Hospitalization in previous year**	8 (18%)	6 (26%)	0.52
**Resident in nursing home**	4 (9%)	1 (4.5%)	0.65
**Comorbidities**			
**Diabetes mellitus**	19 (44%)	3 (13.3%)	0.01
**Cancer**	4 (9.3%)	2 (9%)	1
**Cirrhosis**	2 (4.6%)	0 (0%)	0.54
**Hemodialysis**	0 (0%)	0 (0%)	NA
**Immunosuppression**	1 (2.3%)	0 (0%)	0.47
**Sphincterotomy performed during ERCP**	32 (74.4%)	20 (90%)	0.19
**Common bile duct diameter (mean, cm)**	1.3	1.2	0.78
**Patient outcomes**			
**Hospitalization length (days, mean)**	7.8	5.9	0.09
**In-hospital mortality**	1 (2.3%)	0 (0%)	0.47
**Required Intubation**	1 (2.3%)	0 (0%)	0.47
**Loss of consciousness**	1 (2.3%)	0 (0%)	0.47
**Hypotension requiring vasopressors**	1 (2.3%)	0 (0%)	0.47
**30-day re-hospitalizations**	6 (13.9%)	4 (18.2%)	0.72
**30-day mortality**	0 (0%)	1 (4.5%)	0.15

**Table 5 diagnostics-15-00695-t005:** Multivariate logistic regression analysis of significant independent factors influencing bile culture positivity.

Variable	Coefficient (β)	p-Value	Significance
**Intercept (Constant)**	−1.5298	0.205	Not significant
**Age**	0.0196	0.299	Not significant
**Diabetes mellitus**	1.1394	0.125	Not significant
**Cholangitis severity**	0.7633	0.268	Not significant

**Table 6 diagnostics-15-00695-t006:** Comparison of characteristics and outcomes of patients with cholangitis with appropriate or inappropriate empiric antibiotic coverage.

	Appropriate Antibiotics *n* = 17	Inappropriate Antibiotics *n* = 26	*p*-Value
**Age (years median)**	74	82	0.08
**Female**	11 (64.7%)	15 (57.7%)	0.64
**Days before ERCP (median)**	1	3	0.31
**Received antibiotics before ERCP**	14 (82.3%)	26 (100%)	0.055
**Hospitalization in previous year**	3 (17.6%)	5 (19.2%)	0.89
**Resident in nursing home**	2 (11.7%)	2 (7.7%)	0.67
**Comorbidities**			
**Diabetes mellitus**	8 (47%)	11 (42.3%)	0.76
**Cancer**	3 (17.6%)	1 (3.84%)	0.12
**Cirrhosis**	1 (5.8%)	1 (3.84%)	0.75
**Hemodialysis**	0 (0%)	0 (0%)	Na
**Immunosuppression**	0 (0%)	1 (3.84%)	0.41
**Sphincterotomy performed during ERCP**	10 (58.8%)	22 (84.6%)	0.058
**Common bile duct diameter (mean, cm)**	1.1	1.23	0.23
**Patient outcomes**			
**Hospitalization length (days, mean)**	7.8	7.9	0.95
**In-hospital mortality**	1 (5.8%)	0 (0%)	0.21
**Required Intubation**	0 (0%)	1 (3.8%)	0.41
**Loss of consciousness**	1 (5.8%)	0 (0%)	0.21
**Hypotension requiring vasopressors**	1 (5.8%)	0 (0%)	0.21
**30-day re-hospitalizations**	3 (17.6%)	3 (3.8%)	0.57
**30-day mortality**	0 (0%)	0 (0%)	NA

## Data Availability

This study’s data will be available by contacting the corresponding author upon request.

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
