# Peer review of "Do the Results of Bile Cultures Affect the Outcomes of Patients with Mild-to-Moderate Ascending Cholangitis? A Single Center Prospective Study"

_diagnostics, 2025, doi:10.3390/diagnostics15060695_

Round 1
Reviewer 1 Report
Comments and Suggestions for Authors
This is an interesting paper comparing bile culture positive and negative cases of acute cholangitis. A limitation is that it did not include cases of severe cholangitis. Please indicate the number of cases with cholangitis severity (mild or moderate) in bile culture positive and negative cases and investigate the impact of severity on bile culture collection. Also, please investigate factors influencing bile culture collection using univariate and multivariate analyses.
Comments on the Quality of English LanguageThe English could be improved to more clearly express the research.
Author Response
We thank the reviewers for their comments and suggestions. We have worked to address all of these issues and we feel that the manuscript is now much improved.
In addition, the manuscript has been edited by one of the authors, Dr. Cohen, who is a native English speaker. We feel that these changes have improved the language and flow of the paper.
Reviewer 1
This is an interesting paper comparing bile culture positive and negative cases of acute cholangitis.
Comment 1: A limitation is that it did not include cases of severe cholangitis.
Answer 1: Thank you for this comment. We admit that this is a limitation and was probably a mistake on our part. We have added further clarification of this matter in the "Study Design" section and the “Limitations” paragraph. We explained that the decision to exclude severe cases when writing the study protocol was based on ethical considerations and the desire to ensure the validity of the analysis.
Comment 2: Please indicate the number of cases with cholangitis severity (mild or moderate) in bile culture positive and negative cases and investigate the impact of severity on bile culture collection. Also, please investigate factors influencing bile culture collection using univariate and multivariate analyses.
Answer 2: Thank you for these important suggestions. We have divided the cases based on severity (see Table 4) and found that moderate cases were significantly more likely to have positive blood cultures than mild cases on univariate analysis. We also performed a multivariate logistic regression analysis, as recommended, including age, diabetes, and cholangitis severity. This analysis showed showed that none of these variables remained significant when evaluated together (new Table 5) likely due to overlapping influences between the variables.
Reviewer 2 Report
Comments and Suggestions for Authors
The study investigates the impact of bile cultures on the outcomes after ERCP procedures. The study highlights that biliary decompression is essential to treat patients with ascending cholangitis, no matter what germ is identified in bile cultures. The study is well-designed and well-presented, the methods are correctly used, and the results sustain the conclusions. Although there is no novelty to diagnosing, pathogenesis, or managing such pathology in the present paper, it may add value to the current literature. Nevertheless, there are particular situations in which immunosuppressive patients with multiple hospitalizations may come into contact with nosocomial multi-resistant pathogens, and bile culture and antibiogram might be essential for increasing the chances for a favorable outcome or to guide the antibiotics treatment in case of no response to initial therapy. Several issues should be addressed before acceptance is considered:
Major issues:
It is unclear why the patients with severe cholangitis were excluded. This is an essential limitation of the study because these patients are the most important due to their poor prognosis. The inability to sign the consent form does not stand because they were submitted to an ERCP.
As Table 5 shows, a few differences were observed in factors that might influence the outcomes in the group of patients with appropriate vs. inappropriate antibiotics. However, statistical significance was not reached. The relatively low number of patients in each group might be a significant drawback and limitation and might influence the results and conclusions of the present study.
No clear recommendations for clinical decision-making are provided based on the results of the present study.
Minor issues:
Please consider replacing “all P = NS” with “ p-values ≥ the lowest p-value.” (i.e., p-values ≥ 0.320, ns).
In line 71, please replace “Geroge“ with “George.”
Please consistently provide data such as the number of patients (%) throughout the manuscript.
Comments on the Quality of English LanguageA native English speaker should revise the manuscript to improve fluency and correct a few spelling, grammar, or punctuation errors.
Author Response
We thank the reviewers for their comments and suggestions. We have worked to address all of these issues and we feel that the manuscript is now much improved.
In addition, the manuscript has been edited by one of the authors, Dr. Cohen, who is a native English speaker. We feel that these changes have improved the language and flow of the paper.
Reviewer 2
The study investigates the impact of bile cultures on the outcomes after ERCP procedures. The study highlights that biliary decompression is essential to treat patients with ascending cholangitis, no matter what germ is identified in bile cultures. The study is well-designed and well-presented, the methods are correctly used, and the results sustain the conclusions. Although there is no novelty to diagnosing, pathogenesis, or managing such pathology in the present paper, it may add value to the current literature. Nevertheless, there are particular situations in which immunosuppressive patients with multiple hospitalizations may come into contact with nosocomial multi-resistant pathogens, and bile culture and antibiogram might be essential for increasing the chances for a favorable outcome or to guide the antibiotics treatment in case of no response to initial therapy. Several issues should be addressed before acceptance is considered:
Comment 1: It is unclear why the patients with severe cholangitis were excluded. This is an essential limitation of the study because these patients are the most important due to their poor prognosis. The inability to sign the consent form does not stand because they were submitted to an ERCP.
Answer 1: Thank you for highlighting this limitation. When designing this prospective study, we were concerned about issues of informed consent and patient autonomy. While patients provide consent for ERCP procedures, this consent is strictly for clinical purposes. Research participation requires a separate informed consent process, which presents unique ethical challenges, particularly in critically ill patients that often are not capable of providing consent on their own. In severe cholangitis, patients require urgent intervention, and their condition may impair their ability to provide truly informed consent. Additionally, surrogate consent in the emergency settings can be complex and inconsistent across institutions. To uphold ethical integrity and avoid selection bias, we made the decision to focus only on mild and moderate cases, where informed consent could be obtained more easily. In hind sight, we acknowledge that this may have been a mistake.
However, excluding the severe cases may also enhance the validity of our analysis. As the reviewers rightly point out, the management of severe cholangitis typically necessitates immediate broad-spectrum empiric antibiotics and early drainage, regardless of bile culture results. Including these cases could introduce variability and confound the impact of antibiotic appropriateness on clinical outcomes. By focusing on mild and moderate cases—where antibiotic selection can be more directly influenced by bile culture results—our study provides a clearer and more clinically relevant assessment of how culture-guided therapy affects hospitalization outcomes.
Further clarifications of this issue were added to both the "study design" section and the “limitations” paragraph and are marked in yellow.
Comment 2: As Table 5 shows, a few differences were observed in factors that might influence the outcomes in the group of patients with appropriate vs. inappropriate antibiotics. However, statistical significance was not reached. The relatively low number of patients in each group might be a significant drawback and limitation and might influence the results and conclusions of the present study.
Answer 2: Thank you for this observation. As noted, the lack of statistical significance in Table 5 (now changed to Table 6) may indeed be influenced by the limited sample size, which can reduce the power of the analysis and the ability to detect meaningful differences. We performed 160 ERCP procedures, with 65 patients undergoing the procedure for cholangitis; however, only 43 of these patients had a positive bile culture, and statistical analysis was conducted on this subset. The relatively low number of positive cultures is partly due to the difficulty in predicting the prevalence of positive cultures in cholangitis patients, as there is limited data in the literature to guide such estimates. This made it challenging to estimate the required sample size for the study. While our findings should be interpreted with caution due to this limitation, we believe that the observed trends still provide valuable preliminary insights into the factors that might influence outcomes in patients with appropriate versus inappropriate antibiotic treatment. We acknowledge this limitation and suggest that future studies with larger sample sizes could further clarify these associations.
Comment 3: No clear recommendations for clinical decision-making are provided based on the results of the present study.
Answer 3: This is a correct observation. As mentioned above, while we believe our study provides valuable preliminary insights into the impact of antibiotic appropriateness on outcomes in cholangitis patients, we acknowledge that the current study's limitations, such as the small sample size, prevent us from making definitive recommendations. We agree that larger studies with more robust data are needed to provide clearer, evidence-based tools for clinical decision-making in this area. We hope that our findings can serve as a foundation for future research.
Comment 4: Please consider replacing “all P = NS” with “p-values ≥ the lowest p-value.” (i.e., p-values ≥ 0.320, ns).
Answer 4: We have made these changes as recommended.
Comment 5: In line 71, please replace “Geroge“ with “George.”
Answer 5: We have made this correction.
Comment 6: Please consistently provide data such as the number of patients (%) throughout the manuscript.
Answer 6: We have made the necessary corrections to ensure that the number of patients and percentages are consistently provided throughout the manuscript.
Round 2
Reviewer 1 Report
Comments and Suggestions for Authors
Because this study excluded cases of severe cholangitis, the title and text should state "mild to moderate acute cholangitis."
Author Response
Comment 1: Because this study excluded cases of severe cholangitis, the title and text should state "mild to moderate acute cholangitis."
Answer 1: The corrections to the title and text were made.
Reviewer 2 Report
Comments and Suggestions for Authors
The authors correctly addressed all significant concerns raised by the reviewers. Furthermore, a pertinent explanation was provided for a few suggestions that could not be addressed and stated as limitations of the study.
Author Response
Comment 1: The authors correctly addressed all significant concerns raised by the reviewers. Furthermore, a pertinent explanation was provided for a few suggestions that could not be addressed and stated as limitations of the study.
Answer 1: Thank you very much. The final version has been uploaded.